# Quantifying patterns of alcohol consumption and its effects on health and wellbeing among BaYaka hunter-gatherers: A mixed-methods cross-sectional study

Jessica K. Knight[1,2☯], Gul Deniz Salali[3], Gaurav Sikka[3], Inez Derkx[3,4], Sarai M. Keestra[3,5], Nikhil Chaudhary[1,3☯]*

**1** Department of Archaeology, Leverhulme Centre for Human Evolutionary Studies, University of Cambridge, Cambridge, United Kingdom, **2** University of Cambridge School of Clinical Medicine, Cambridge Biomedical Campus, Cambridge, United Kingdom, **3** Department of Anthropology, University College London, London, United Kingdom, **4** Department of Anthropology, University of Zurich, Zürich, Switzerland, **5** Department of Anthropology, Durham University, Durham, United Kingdom

☯ These authors contributed equally to this work.
* nc542@cam.ac.uk

**Data Availability Statement:** All relevant data are within the manuscript and its Supporting information files.

## Abstract

Ethnographers frequently allude to alcoholism and related harms in Indigenous hunter-gatherer communities, but very few studies have quantified patterns of alcohol consumption or its health and social impacts. We present a case study of the Mbendjele BaYaka, a Congolese population undergoing socioeconomic transition. 83 adults answered questions about their frequency and quantity of alcohol consumption, underwent biometric measurements and reported whether they were currently experiencing a cough or diarrhoea; 56 participated in structured interviews about their experiences with alcohol. Based on WHO standards, we found 44.3% of the full sample, and 51.5% of drinkers (excluding abstainers), had a hazardous volume of alcohol consumption; and 35.1% of the full sample, and 40.9% of drinkers, engaged in heavy episodic drinking; consumption habits varied with sex and age. Total weekly consumption was a positive predictor of blood pressure and the likelihood of experiencing diarrhoea; associations with other biometric variables were not statistically significant. Interview responses indicated numerous other economic, mental and physical health harms of alcohol use, the prevalence of which demonstrate some variability between forest camps and permanent village settlements. These include high rates of drinking during pregnancy and breastfeeding (~40%); frequent alcohol-induced violence; and considerable exchange of foraged foods and engagement in exploitative labour activities to acquire alcohol or repay associated debts. Our findings demonstrate the prevalence of hazardous alcohol consumption among transitioning hunter-gatherers is higher than other segments of the Congolese population and indicate negative impacts on health and wellbeing, highlighting an urgent need for targeted public health interventions.

**Funding:** This work was funded by UCL Grand Challenge of Global Health small grant to GDS (https://www.ucl.ac.uk/grand-challenges/global-health) and British Academy research grant SRG \171409 to GDS (https://www.thebritishacademy.ac.uk). The funders had no role in study design, data collection and analysis, decision to publish, or preparation of the manuscript.

**Competing interests:** The authors have declared that no competing interests exist.

## Introduction

Drinking alcohol is associated with pleasure and relaxation worldwide, and many people consume alcohol without experiencing major adverse effects. However, a proportion of people who drink experience harms due to the toxic, psychoactive, and dependence-producing properties of alcohol. Problems associated with alcohol use have been identified as a major health risk facing Indigenous communities worldwide [1]. Ethnographers and other researchers have alluded to the pervasiveness of alcoholism in numerous traditionally hunting and gathering African societies, including the Kalahari San as well as Central African "Pygmy" populations such as the Baka from Cameroon and the Twa of Rwanda and Burundi [2–5]. In each of these populations, researchers have suggested that high levels of alcohol consumption have contributed to a deterioration of physical health [6, 7]; social problems of increased violence, domestic abuse, delinquency and rape [2, 8]; and economic difficulties stemming from engagement in exploitative labour activities to purchase alcohol, or in some cases, be paid in alcohol directly [8–10]. These reports usually assert that such alcohol-related problems are principally a consequence of the social, political and economic marginalisation that is concomitant with socio-economic transition and sedenterisation [11].

Despite these concerning depictions of alcoholism and its array of harmful consequences, research focusing on Indigenous alcohol use outside high-income countries is extremely rare. Quantitative data on consumption patterns and health consequences of alcohol are virtually non-existent for African hunter-gatherers, this is likely due to the difficulty in collecting data from populations residing in remote areas with dispersed communities and some degree of nomadism [6]. An exception is a study that found raised serum gamma-glutamyl transferase—a biomarker of heavy drinking—in 30% of men and 11% of women in San communities, which were demonstrated to contribute to thiamine deficiency in the sample [12]. However, other suggested links with morbidity, such as hypertension among the Bakola of Cameroon, remain speculative [3, 4]. The more substantial research among Australian Aborigines suggest that if the process of socio-economic transition among Indigenous populations in Africa mirrors that of Australia, alcohol-related health problems are likely to be vast and severe [1]. Accordingly, public health scholars have stressed the urgent need for systematic data collection and focussed original research [1, 13].

Here we present data on alcohol consumption and a preliminary investigation into its effects on health and wellbeing among the Mbendjele of Congo. They are one of numerous "Pygmy" hunter-gatherer populations living in the rainforests of Central Africa and an ethno-linguistic subgroup of the BaYaka, which also includes the Aka, Luma, Mikaya, Ngombe and Baka [14]. Traditionally, they are highly mobile, and rely primarily on subsistence foraging. However, increased logging and "fortress" conservation means many Mbendjele have moved to more permanent settlements, living alongside Bantu people and participating in alternative economic activities [15]. These changes occur on a backdrop of discrimination against the BaYaka by neighbouring non-Pygmy groups [16]. As in other traditionally hunting and gathering populations, the process of socio-economic transition has been arduous for the Mbendjele and increased alcoholism described. For example, Lewis writes that since beginning his fieldwork among the Mbendjele in 1994, "a society of active, well-fed hunter-gatherers. . . [have] become poorly nourished agricultural day labourers, or clandestine hunter-gatherers sedentarised by terror. . .and alcoholised to encourage debt bondage and pass the time [15, p7]."

An increasing variety of alcoholic drinks have been reported in BaYaka communities (summarised in Table 1), as well as concerns regarding their methanol content, which may cause disorders of the nervous system and cancers [3]. However, only one study has attempted to

**Table 1. Types and prices of alcoholic drinks previously reported in BaYaka communities.**

| Name(s) | Description and reported %ABV | Source | Reported Price (100CFA = $0.16USD)[a] |
|---|---|---|---|
| *Ngolo'ngolo* (also *lotoko*, *menyoke*) | Home-distilled spirit made from maize and cassava; 30–45% ABV[b] or up to 80% ABV[c] | Farmers | 1000CFA/litre; 100CFA/100ml glass; one day's labour/litre |
| *Molenge* (also *mbila*, *mbolo*, *menyok me leer*) | Wine produced from local raffia/oil palm; 3–4% ABV though some batches stronger[b] | Farmers | 500CFA/bottle[d] |
| 'Whisky' (also *nofia*, brand names e.g. 'King Arthur', 'Officer') | Commercially manufactured spirit widely available since late 2000s; 40–50% ABV; sold in 40-50ml sachets; banned by government of Cameroon[b] | Merchants; local bars | 50–300CFA/Sachet[b,d] |
| *Njàmbù* | Honey wine, infrequently consumed[b] | Unknown | Unknown |
| Beer | Commercially manufactured bottled beer, used as a status symbol by Baka[e] | Merchants; local bars | 800CFA/bottle[d] |
| *Vinesol* | Commercially manufactured wine in paper packs[b] | Merchants; local bars | Unknown; generally too expensive for Baka[b] |
| Pastis (also *Ricard*) | Commercially produced aniseed spirit, 40–45%ABV[f] | Farmers; merchants | Unknown |

a. Exchange rate on 19/04/21.

b [10]

c [17]

d [3]

e [18]

f [19]

quantify alcohol use in a BaYaka society—the Baka of Cameroon—and these data are limited by a sample size of 12 individuals [10]. Thus, whilst considerable ethnographic work is suggestive of potential adverse effects of alcohol consumption in BaYaka people, current research is inadequate in its scope and methods. Given the combined pressures of deprivation, exploitation and rapid acculturation which BaYaka populations are currently facing, this research is essential to provide an evidence base for programmes to improve health and social outcomes. In this study we i) estimate the prevalence of hazardous drinking and how it varies with age and sex; ii) examine the association between individual consumption levels and various biometric measures and health indicators; and iii) present and analyse some of the Mbendjele's perspectives, experiences and beliefs related to alcohol.

## Methods

### Study population

The Mbendjele reside in the forests of Congo and Central African Republic. Traditionally, they are mobile, living in camps of 10–60 individuals [17]. Their subsistence relies on hunting and gathering; and some exchange of forest products with neighbouring Bantu people for alcohol, cigarettes, and manioc [20]. They are predominantly serially monogamous, though there is some incidence of polygyny [21]. Ritual is a core component of Mbendjele life, all members of camp participate when *massana* (ritual singing and dancing) occur, many *massana* are shared with other BaYaka groups. *Massana* can last anywhere from a few hours to a few days, and it is only when the singing, dancing and clapping are vigorous and beautiful enough that the *mokondi* (spirits of the forest) arrive [17]. In most cases, alcohol is a required ingredient for *massana* to take place, partly because the Mbendjele perceive it to provide the necessary strength, stamina, and joy.

Like most hunter-gatherer populations worldwide, the Mbendjele are undergoing a socioeconomic transition. Some communities have become largely sedentary, engage in

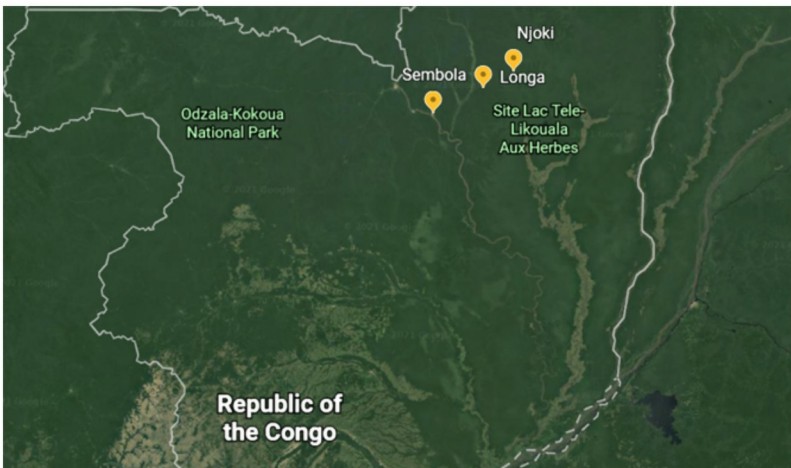

**Fig 1. Map of study camps, constructed using Google Earth.**

supplementary economic activities such as wage labour and horticulture, have greater market access and reduced traditional ecological knowledge [22, 23]. We studied three Mbendjele camps—Longa, Njoki and Sembola—situated in Congo's Ndoki forest (see Fig 1 for map). The socioeconomic changes mentioned above were particularly visible in the large permanent settlement of Sembola, which is located in the logging town of Pokola and walking distance from a market and hospital. Up to several hundred Mbendjele stay in Sembola at certain times of the year, some live there permanently whilst others partition their time between this settlement and the forest depending on economic opportunities and needs [23]. Longa and Njoki are more isolated in the forest but still situated beside a logging road. Camp sizes at these two locations are constantly changing as Mbendjele come and go. At the time of this study, there were approximately 60 Mbendjele adults in Longa and 25 in Njoki. Additionally, there were two Bantu families staying in Longa, and in turn some trade and labour opportunities for the Mbendjele living there, as well as alcohol for sale. At Njoki, there were no Bantu residents, but alcohol and other goods could be acquired if a vehicle passed by on the logging road or if someone travelled to purchase it.

## Data collection

During data collection in Longa and Njoki we stayed in each camp for 2–3 weeks, whereas when collecting data from Sembola we stayed in a nearby hotel. Informed consent was obtained by translators reading an information sheet in Mbendjele and asking for a confirmatory signature (thumbprint or pen mark) from those willing to participate. This research had ethical approval from the UCL ethics board and permission was granted by Congo's Ministry of Scientific Research.

Data on quantities of alcohol consumed and biometric measurements were collected from all consenting adults in Longa and Njoki (n = 83; 46 women). They were asked:

1. How many times per week do you drink alcohol?

2. When you drink alcohol, how many cups do you drink per time?

3. What type of alcohol do you drink?

We chose the above questions rather than timeline follow-back methods in which participants recount their consumption over the last two weeks. This is because Mbendjele alcohol consumption is not consistent across time, rather it is determined by access, ability to purchase and the timing of ritual events. Therefore, asking a general question rather than focussing on the previous two weeks is likely to elicit average habits and provide a more representative depiction of consumption patterns.

The physiological variables measured were blood pressure (mmHg, maximum and minimum values), heart rate, haemoglobin (g/L), glycated haemoglobin HbA1c (mmol/mol), total white blood cell count ($10^9$/L) and BMI. These measurements were taken by GS who is a qualified NHS physician. Blood pressure and heart rate were recorded using an omron blood pressure monitor. Haemoglobin and HbA1c were recorded using a hemocue HbA1c-501 system. White blood cell count was recorded using a hemocue WBC-Diff system. Height and weight (to calculate BMI) were measured with a seca portable stadiometer and seca electronic floor scale respectively. Measurements were taken as privately as possible either in a hut or an opaque shelter constructed using tarpaulins and stands. We also asked each participant if they were currently experiencing diarrhoea or a cough, since these are the most common health problems reported by the Mbendjele, and respiratory tract infections and diarrhoeal diseases have been highlighted as a major cause of morbidity and mortality in Indigenous populations and hunter-gatherers [1, 24].

Structured interviews focused on drink preferences, acquisition of alcohol, drinking during pregnancy/breastfeeding, reasons for drinking and beliefs about alcohol. These were carried out in Sembola and Longa (n = 56; 24 women), see S1 Table in S1 File for a complete list of questions. The mixed-methods approach was designed to ensure Mbendjele people's perspectives and experiences of alcohol use formed a central part of the study. Interview questions were informed by preliminary discussions about alcohol use with elders in the community and some questions were added during the study process based on themes introduced by participants. The sampling strategy for interviews was opportunistic. Interviews were conducted in Mbendjele by a French-speaking researcher who was accompanied by a translator who speaks French but is Mbendjele himself; they all took place in private. Sample characteristics for both phases of the study are shown in Table 2.

## Analysis

All data were anonymised prior to analysis. To permit comparisons with other populations, measurements were converted from Mbendjele cups to metric units. In accordance with our own observations and estimates among the Baka [10], a cup was estimated to measure 0.1L. WHO definitions of harmful alcohol consumption are [25]:

- Regular consumption of a hazardous volume of alcohol, averaging more than two drinks—over 24g ethanol/day.

- Heavy episodic drinking (HED), defined as sometimes consuming about five or more drinks—roughly 60g of ethanol—on a single occasion.

The frequencies of different types of alcohol reported were multiplied by the lowest previously reported percentage alcohol by volume (%ABV) to estimate a conservative weighted mean ethanol content. We applied this weighted mean ethanol content per drink to the whole sample rather than calculating ethanol content per drink for each individual based on the specific drink reported by each participant. This is because the Mbendjele drink together in groups and will usually drink the same drink as the rest of the party on any given occasion. Whilst it may be unrelatable in industrialised cultures, our understanding is that individuals

**Table 2. Sample characteristics for quantitative analyses and structured interviews.**

| Study | | Biometrics | |
|---|---|---|---|
| **Camp** | Longa | Njoki | Total* |
| **Sex** | | | |
| males | 26 (47.3%) | 9 (34.6%) | 37 (44.6%) |
| females | 29 (52.7%) | 17 (65.4%) | 46 (55.4%) |
| **Age** | | | |
| 15–29 | 17 (31%) | 7 (26.9%) | 25 (30.1%) |
| 30–49 | 13 (23.5%) | 13 (50%) | 26 (31.3%) |
| 50+ | 14 (25.5%) | 6 (23.1%) | 20 (24.1%) |
| N.A. | 11 (20%) | 0 (0%) | 12 (14.5%) |
| **mean (s.d.)** | 42.3 (19.6) | 38.4 (14.4) | 40.6 (17.8) |
| **Study** | | **Interviews** | |
| **Camp** | Longa | Sembola | Total |
| **Sex** | | | |
| males | 12 (48%) | 20 (64.5%) | 32 (57.1%) |
| females | 13 (52%) | 11 (35.5%) | 24 (42.9%) |

* Two individuals from another camp (Bilikombo) also participated in the biometric phase of the study; therefore, the total column contains an extra two data points.

do not decide whether to drink or not on a given occasion based on which drinks are available, choice of drink is a secondary decision and usually determined by availability. It is likely the question was interpreted as which drink would the participant choose if all drinks were available, and it is certainly not the case that individuals only consume the drink they reported in their response. Estimating average ethanol content per drink for each individual based on each participant's specific drinking history would be ideal if it were possible to accurately obtain this data. However, we do not believe this would change our results very much since 90% of responses listed either lotoko, pastis or apollon, which are all spirits with similar %ABV (~40%), and the other drinks listed are less commonly available.

The sample weighted mean ethanol content per drink and average volume per cup (ml) was then converted in to grams of ethanol/per cup. This was then used to estimate rates of hazardous drinking, HED, and per capita alcohol consumption. Chi-squared tests were used to test for sex differences in the incidence of hazardous drinking, HED and abstention.

Multiple regression was used to investigate the effects of age and sex on i) frequency of alcohol consumption per week, ii) amount of alcohol consumed per occasion, and iii) total weekly alcohol consumption. Potential quadratic relationships with age were accounted for by including an $age^2$ term if this increased adjusted $R^2$. Estimated ages had already been produced for a large sample of this study population by creating relative age lists with the community and assigning age ranges for individuals based on dental development, sibling birth orders and triangulating their birth years with known events such as the construction of a logging road. Final age estimates were then obtained by integrating this information using a Gibbs sampling framework. Detailed information and validation of this method can be found in [26].

We also employed multiple regression to examine the effect of total weekly alcohol consumption on i) systolic and diastolic blood pressures, ii) haemoglobin levels, iii) white blood cell count, iv) BMI, v) glycated haemoglobin, and vi) heart rate. Age and sex were included as controls in each of these models, and interactions between weekly consumption and age or sex

were included if they increased the adjusted $R^2$. Potential quadratic relationships with age were accounted for by including an age$^2$ term if this increased adjusted $R^2$.

We used logistic regression to examine the effect of total weekly alcohol consumption on the odds of reporting current i) diarrhoea and ii) cough, controlling for age and sex.

Answers to the structured interviews were coded with categories developed inductively from themes present in participants' responses. Responses were coded in more than one category if multiple themes were present. Following the initial open coding, data were iteratively re-analysed to ensure the validity of the proposed categories, see S1 File for details of coded categories. Chi-squared tests were used to test for differences in responses by sex and camp.

## Results

### Quantifying alcohol consumption

The consumption data was pooled as there were no significant differences between consumption in Longa and Njoki. Participants reported drinking alcohol a mean frequency of 2.4 days per week (s.d. = 2.1, male mean = 2.5, female mean = 2.3), with a mean volume of 2.7 cups per occasion (s.d. = 2.3, male mean = 3.9, female mean = 2.0). Mean alcohol consumption per week was 7.4 cups (s.d. = 8.7, male mean = 9.9 cups, female mean = 5.6 cups), see Fig 2 for summary.

If we exclude abstainers from our calculations, consumption patterns for alcohol drinkers are as follows: the mean frequency of drinking is 2.8 days per week (s.d. = 2.0, male mean = 2.9, female mean = 2.7); with a mean volume of 3.1 cups per occasion (s.d. = 2.2, male mean = 4.2, female mean = 2.3); and a mean weekly consumption of 8.7 cups (s.d. = 8.7, male mean = 11.6, female mean = 6.5).

Eighty-two responses to "which type of alcohol do you drink" were given, including lotoko (n = 55), pastis (n = 10), apollon [gin brand] (n = 8), molenge (n = 3), beer (n = 2), and vinesol

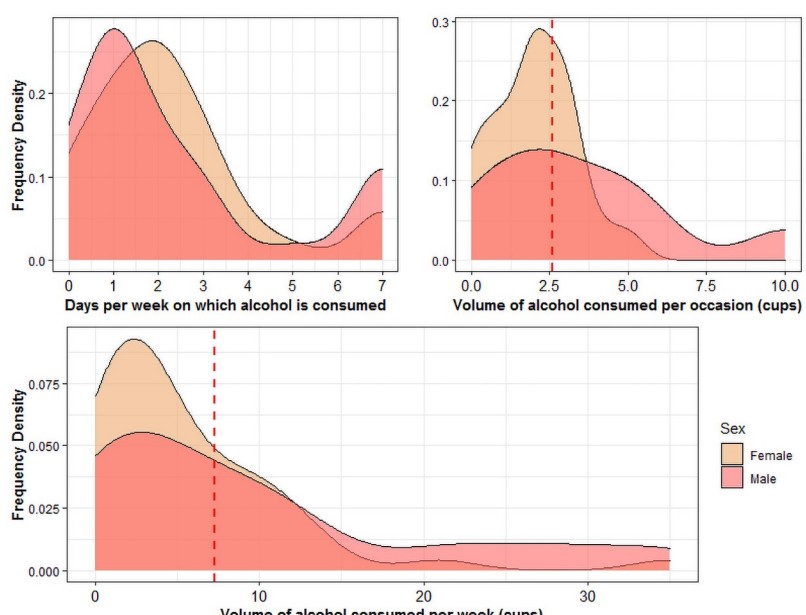

**Fig 2. Frequency density curves showing the distribution of alcohol consumption behaviours.** Dashed red lines indicate estimate of WHO-defined hazardous drinking.

**Table 3. Estimated sample prevalence of different drinking behaviours, using WHO definitions and a conservative estimate of average 29.4% ABV in Mbendjele drinks.**

| | Men | Women | Total | Mean age (years) | Mean quantity/ occasion (cups) | Mean freq. / week (days) | Mean vol. / week (cups) |
|---|---|---|---|---|---|---|---|
| **Volume per occasion (n = 79; 34 men, 45 women)** | | | | | | | |
| Abstention | 5 (14.7%) | 6 (13.3%) | 11 (13.9%) | 23.5 | 0 | 0 | 0 |
| Heavy episodic drinking | 19 (55.8%) | 16 (35.5%) | 35 (44.3%) | 46.1 | 4.5 | 3.1 | 13.1 |
| Non-HED drinking | 10 (29.4%) | 23 (51.1%) | 33 (41.8%) | 36.1 | 1.6 | 2.4 | 4.0 |
| **Volume per week (n = 77; 33 men, 44 women)** | | | | | | | |
| Non-hazardous volume of drinking | 18 (54.5%) | 32 (72.7%) | 50 (64.9%) | 35.6 | 2.1 | 1.7 | 3.3 |
| Hazardous volume of drinking | 15 (45.5%) | 12 (27.3%) | 27 (.35.1%) | 46.9 | 4.6 | 4.2 | 16.4 |

(n = 2). These frequencies were used to estimate a weighted mean ethanol content of 29.4%. Hazardous drinking and HED were therefore considered respectively as:

- Drinking over 7.3 Mbendjele cups of alcohol per week

- Drinking over 2.6 Mbendjele cups of alcohol on a single occasion

The mean total weekly alcohol consumption was used to estimate annual per capita consumption of ethanol, which was 11.4L ethanol per capita (s.d. = 13.2), 15.4L for men and 8.7L for women. Men were more likely to engage in heavy episodic drinking (chi-squared = 3.24, p = 0.072) and overall hazardous drinking (chi-squared = 2.74, p = 0.098), though these results had p-values marginally above the p<0.05 significance level. There was no sex difference in rates of abstention (chi-squared = 0.030, p = 0.862). Notably the mean age of participants who did not drink was significantly younger, by more than 20 years, than both those who drank at hazardous levels (W = 11.5, p = 0.001) and those that engaged in heavy episodic drinking (W = 15, p = 0.001), see Table 3 for summary.

Accordingly, in the regressions predicting total weekly alcohol consumption (cups), sex and age were significant predictors. Men consumed 4.6 cups more per week (p = 0.031), principally due to consuming more cups per occasion (coefficient = 1.64, p = 0.003), rather than drinking more frequently than women (coefficient = 0.22, p = 0.650). There was a quadratic relationship between age and weekly consumption, specifically individuals drink more as they get older, up until old age (slope remains positive until age 57). Age was positively associated with frequency of alcohol consumption (coefficient = 0.047, p = 0.001); however, the relationship between age and amount consumed per occasion was quadratic, again remaining positive until older ages (slope remains positive until age 55.6), see S2 Table in S1 File for full results.

## Alcohol and health

The only biometric variable significantly predicted by total weekly alcohol consumption was systolic blood pressure (coefficient = 1.98, p = 0.026), the relationship with diastolic blood pressure (coefficient = 0.76, p = 0.062) approached statistical significance. Weekly consumption also significantly increased the odds that a participant reported currently experiencing diarrhoea (OR = 1.12, p = 0.008). Total weekly alcohol consumption was not significantly associated with the other biometric measurements or the likelihood of reporting a cough, see S3 Table in S1 File for full results.

In the structured interviews, 39.2% of people reported that they/their partner had drunk alcohol whilst pregnant, and 41.7% whilst breastfeeding. Drinking during breastfeeding was more than five times as common in the camp of Longa than the settlement of Sembola (chi-

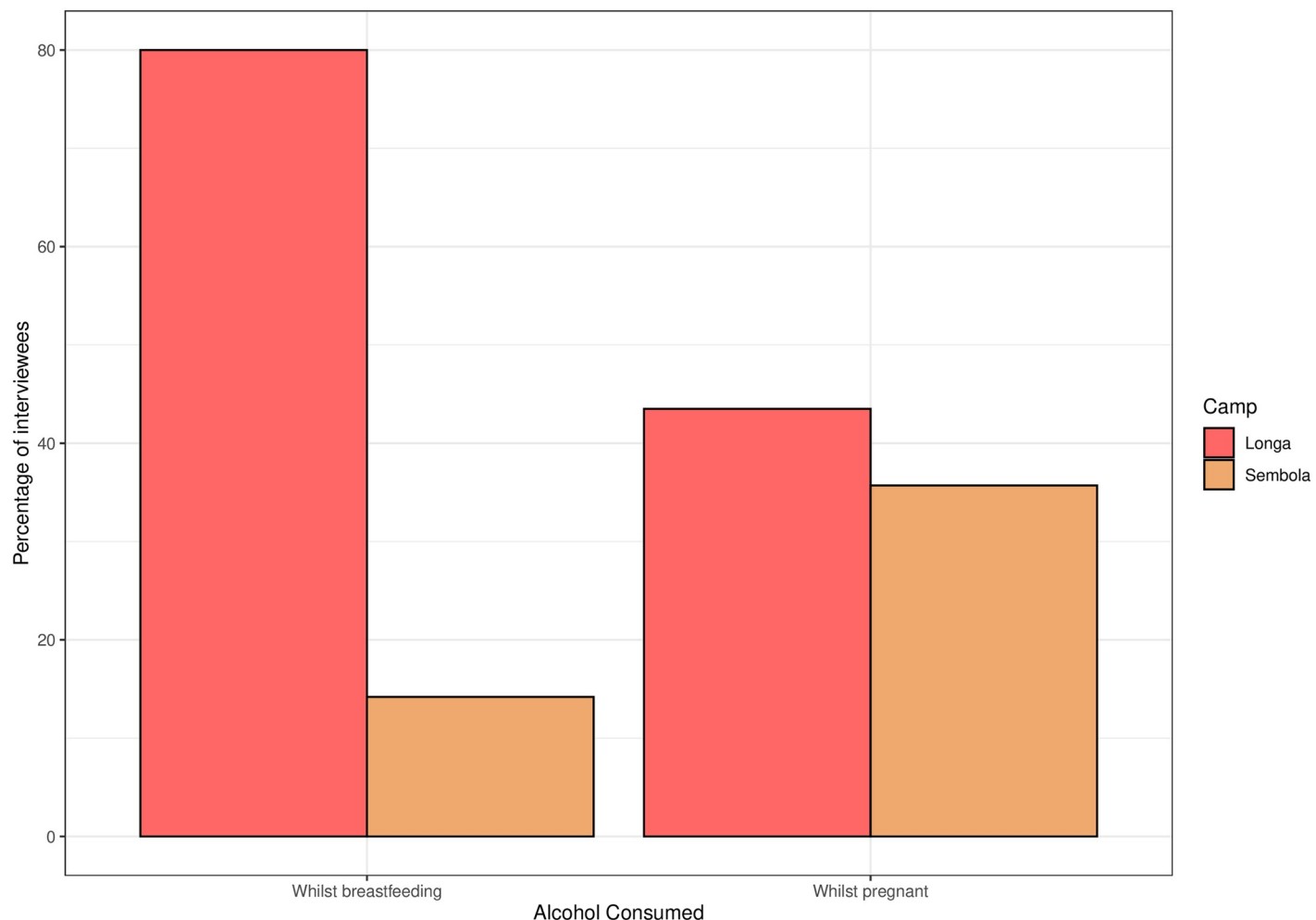

**Fig 3. Percentage of participants reporting that they or their partner consumed alcohol while pregnant or breastfeeding, separated by camp.**

squared = 20.7, p<0.0001), see Fig 3. Some respondents reported beliefs regarding the impact on the fetus of drinking during pregnancy, saying that the child would be born drunk or that alcohol can kill the baby. Dangers to pregnant women from drinking were also mentioned, including risk of rape, violence or accidents. However, others mentioned positive impacts, including giving strength to the child and reducing the pain of labour. It was suggested multiple times that drinking molenge during breastfeeding was beneficial, and that molenge was non-alcoholic. One respondent suggested that drinking vinesol was particularly beneficial, as it could replace the blood lost in delivery.

## Social and economic consequences

The traditional home-brewed beverages were most frequently named as favourite drinks, lotoko (30.6%) and molenge (19.3%), followed by commercially produced beer (17.7%) and pastis (11.3%), the other drinks mentioned were apollon, café rum and vinesol. Commercially produced drinks were more popular in Sembola than Longa, but this difference was not statistically significant (chi-squared = 1.04, p = 0.307).

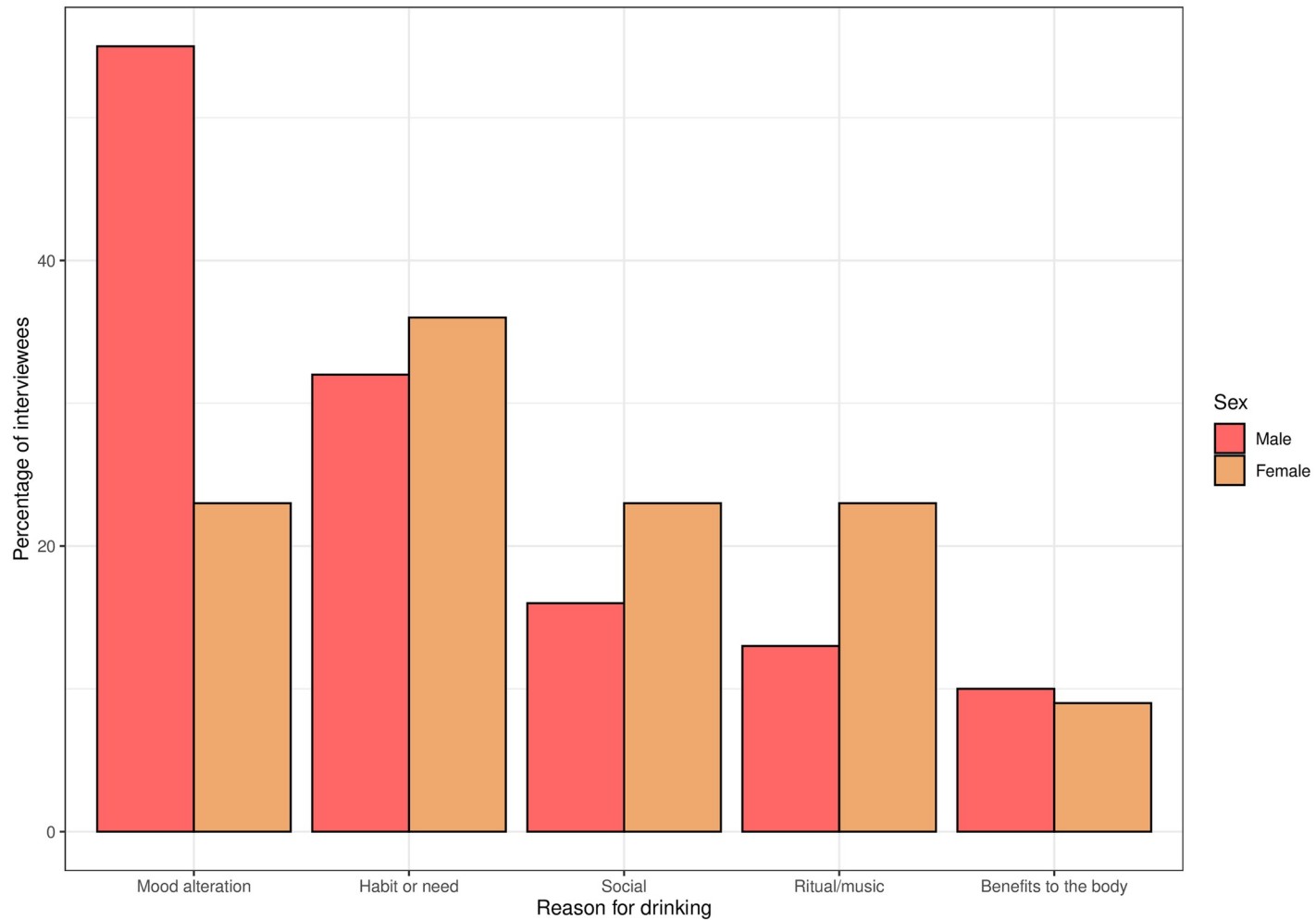

**Fig 4. Bar chart of participants' stated reasons for drinking, separated by sex.**

Two-thirds of interviewees purchased their most recent alcoholic drink from a Bantu-run stall; others obtained alcohol from Bantu people in exchange for labour (17.6%) or from a friend/family member (15.7%). Among those who bought alcohol, the money was most often earned by selling foraged foods (33.3%) or manual labour for the Bantu (33.3%), though some people in Sembola earned money from other sources e.g. work for the forestry company. 60.9% of participants asked stated they had entered into debts with Bantu people to buy alcohol before, in a majority of these cases interest was applied.

The most frequent reasons for drinking alcohol are summarised in Fig 4. 41.5% of respondents mentioned mood alteration and 34.0% mentioned habit/need. Social, ritual and health benefits were mentioned by 18.8%, 17.0% and 9.4% of respondents respectively. Men were more likely than women to state mood alteration as a reason for drinking (chi-squared = 5.47, p = 0.019), there were no other significant sex differences.

When interviewees were asked whether they thought alcohol causes problems, violence (including domestic violence) was the most frequent response (44.4% of responses), and reported significantly more often in Sembola than Longa (chi-squared = 5.97, p = 0.015). Other problems raised include disorder/disturbance and people talking nonsense; it is

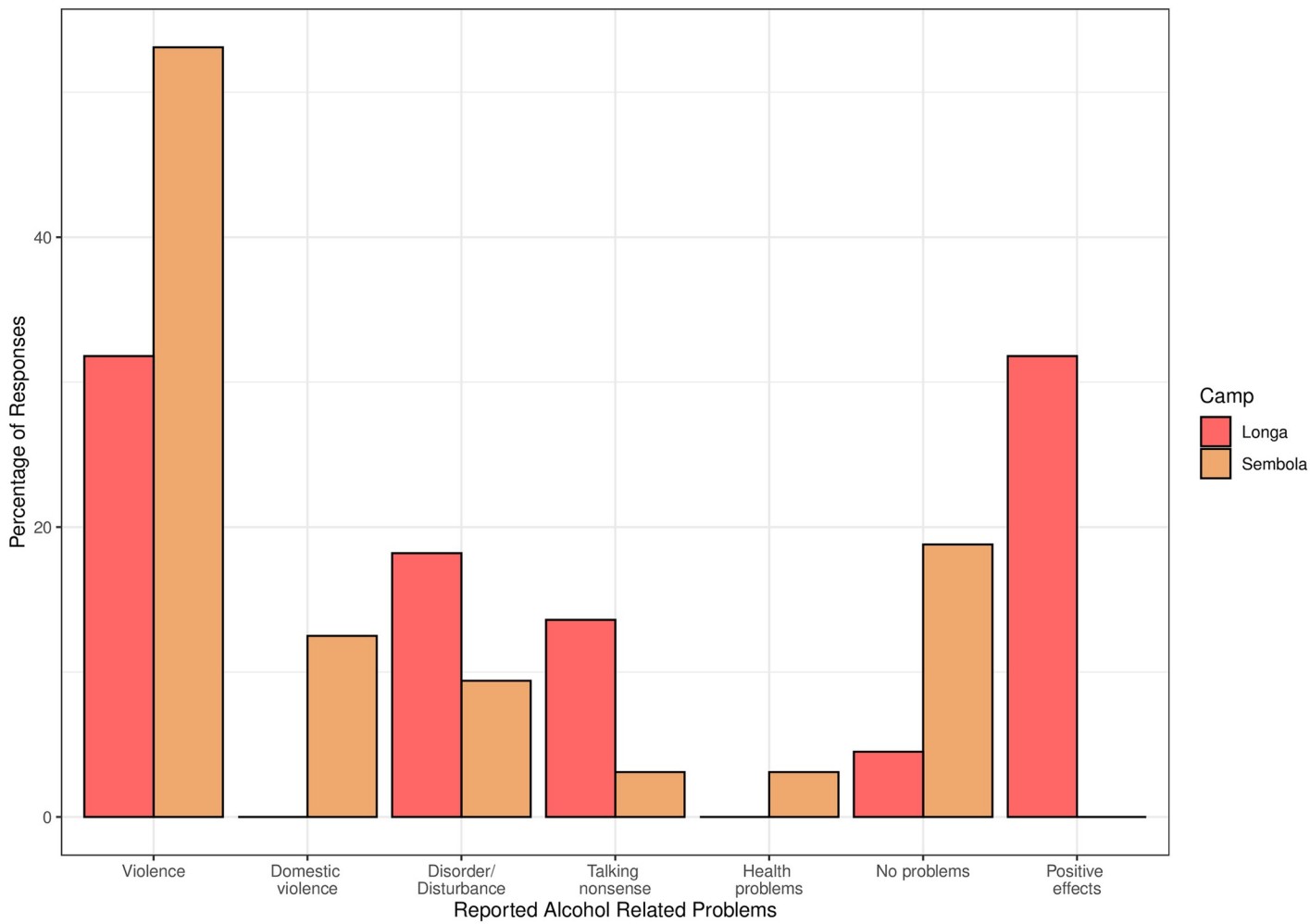

**Fig 5. Bar chart of participants' responses to whether alcohol causes problems, separated by camp.**

noteworthy that only one participant mentioned health problems, see Fig 5. Whilst no interviewees in Sembola alluded to a positive effect of alcohol in response to this question, in Longa 36.4% of responses referred to positive effects, principally that alcohol is "good during music/ dance"; this camp difference is significant (chi-squared = 2.11, p = 0.0006). See S4 Table in S1 File for frequencies of all response themes separated by camp.

## Discussion

### Alcohol consumption

In 2016, 48.3% of the Congolese population over the age of 15 had abstained from alcohol for the past year [25], by contrast, in this sample only 13.9% of people reported abstention. Estimated annual per capita ethanol consumption in this sample was higher than national levels, at 15.4L for men and 8.7L for women compared with nationwide estimates of 12.9L and 2.8L respectively. Consumption quantities of this Mbendjele sample were similar, though slightly higher, to those found in an observational report of 12 Baka [10]. We report a mean of 7.4 cups per week (9.9 cups for males and 5.6 cups for females) and the data from the Baka study

equate to a mean of 6.8 cups per week (9.2 cups for males and 4.4 cups for females). Estimates from this sample show that a substantial proportion of Mbendjele engage in HED, including 55.8% of men and 35.5% of women; similar to nationwide Congolese estimates. It is clear that many Mbendjele drink alcohol frequently, habitually, and in greater quantity than national averages. It is also possible that the per unit effects of alcohol are greater on Mbendjele people, due to their shorter stature [27], Pygmy populations are characterised by an average male height <155cm [28].

Our findings suggest that in Mbendjele communities, older people drink more frequently; and men and older people (other than those aged ~60 and over) drink larger volumes on a single occasion. Interestingly, this positive association with age is at odds with some descriptions of other hunter-gatherer populations, which imply that younger members of the community drink more [2, 29]. This contrast may reflect genuine differences in consumption trends between populations; or may be a consequence of older people's drinking being less attention grabbing and in turn less likely to be noted in the absence of systematic data collection. Hence more research is needed to identify whether the similar reports of alcoholism in transitioning hunter-gatherers actually follow distinct population-specific trends. Our finding of greater consumption per occasion among men than women is more consistent with reported trends of other hunter-gatherer populations [7, 10; but see 30]. It is noteworthy that although men in the sample drank more, the difference from national estimates was greater for Mbendjele women [25]. This may relate to a strong tradition of female autonomy and sex egalitarianism in Mbendjele communities [17, 31], and highlights the need for public health interventions that are specific to the Mbendjele cultural context.

## Physical health

Total weekly alcohol consumption predicted higher blood pressure. These associations are consistent with trends in other populations, including urban Congolese populations, where a 41% prevalence of hypertension has been reported [32]. Only 15.7% of Mbendjele in this sample had a blood pressure >140/90mmHg, closely matching the 15% hypertension reported among the Bakola, which Froment had speculated was a consequence of alcohol consumption [4]. The relatively low prevalence of hypertension despite high alcohol use in these African hunter-gatherers may be due to the rarity of other risk factors such as obesity and low physical activity levels, although smoking is common [25, 33].

We also found that greater consumption substantially increased the odds of a participant reporting they were currently experiencing diarrhoea. Excessive alcohol consumption is known to cause diarrhoea both acutely and chronically via a variety of mechanisms including inhibition of water and sodium absorption, altered gut motility and permeability [34, 35]. The severity and incidence of diarrhoeal diseases are very high among hunter-gatherers, including Central African Pygmies, and exacerbated by increased sedentism [36–39]. Moreover, gastro-intestinal maladies are a major—and in numerous cases, the most common—cause of death in hunter-gatherer populations [24]; thus, hazardous alcohol consumption in these populations is particularly dangerous.

This investigation must be treated as preliminary, and the absence of other effects may be because the study lacked sufficient power. Whilst the sample size of this study is relatively small, achieving sample sizes comparable to those of research conducted with less remote populations is extremely difficult, and we hope this exploratory aspect of our study motivates further quantitative research effort.

## Maternal and child health

39% of respondents reported that either they or their partner had drunk alcohol whilst pregnant. This likely underestimates true rates, as it will not include women who drank alcohol when unaware of their pregnancy. Some suspected cases of fetal alcohol spectrum disorders were observed, though this is yet to be studied systemically. The only published investigation into prenatal alcohol exposure in Congo found that 23.3% of women attending prenatal care in Brazzaville reported consuming alcohol whilst pregnant [40]. Our results therefore indicate that alcohol-related harm to pregnant women and their babies may be higher among BaYaka populations. During ritual singing and dancing, we have also observed adults sometimes feeding infants and young children a small amount of alcohol by dipping their finger in an alcoholic drink and then putting it inside the child's mouth.

Few people interviewed showed an awareness of potential harms to health from alcohol use, and some reported benefits to consuming alcohol whilst pregnant or breastfeeding, such as replacing blood loss. Drinking while breastfeeding was significantly less common in Sembola than Longa suggesting that proximity to a hospital and accessible public health information may influence drinking behaviour. However, it is unclear then why the prevalence of drinking during pregnancy is very similar in both camps. One possibility is that understanding the effects of alcohol on the baby from breastfeeding may be more intuitive because the baby is visibly ingesting fluid, whereas nutrient transfer via the placenta during pregnancy is unobservable. Despite key structural determinants of alcohol use, brief communication of the risks of alcohol can be an effective, low-cost and easy-to-administer intervention, and has been utilised successfully in urban Congolese settings [41]. Our findings suggests that such interventions may be beneficial for Mbendjele people, though their effectiveness among hunter-gatherer societies remains to be demonstrated.

## Mental health

Both phases of the study show a minority of participants whose drinking is consistent with alcohol dependence. Ten people reported drinking alcohol every day, while four reported drinking one litre of strong alcohol at least once a week. During interviews, several people compared alcohol to food, explaining that alcohol was a "need", whilst others reported buying alcohol as soon as they had any money. Some interviewees were more explicit in their psychological dependence, with one man saying that he needed alcohol "to stay sane". The frequent occurrence of borrowing from Bantu people and exchange of food for alcohol also suggest dependence.

Heavy alcohol use can also be linked to low mood, including an increased risk of major depressive disorder [25]. When asked why they drank, some Mbendjele respondents linked their drinking to feelings of sadness, saying that drinking helped them "get rid of bad thoughts", "worries", and "sorrow". Reviews of global Indigenous health describe a high prevalence of depression and suicide in Indigenous groups compared with non-Indigenous populations, which is linked to poverty, urbanisation, and a lack of local autonomy and cultural continuity [42]. Whilst discussions with community elders found that none were aware of any Mbendjele people who had died by suicide, interview responses hint that global patterns of Indigenous socioeconomic transition, substance use and mood disorders may potentially be mirrored in BaYaka people.

Problems with alcohol use are frequently comorbid with other substance use disorders [25], and NGOs have reported an increasing incidence of glue sniffing in central African hunter-gatherer communities [43]. In the sites examined here, we saw a shift from virtually no incidence of glue sniffing in 2014 to frequent use in Sembola in 2018 and limited, but non-zero,

use in Longa. During interviews, participants in this study mentioned using both glue and cannabis. Cannabis use occurs in most Congo Basin hunter-gatherer populations and is inversely associated with intestinal parasite infections among the Aka [44]. Therefore, use of psychoactive substances (in addition to alcohol) in BaYaka communities requires further study to provide effective early intervention when appropriate.

## Social and economic impact

Many Mbendjele people reported enjoying drinking alcohol, and described alcohol as crucial to ritual performances such as *Ejɛngi* and *Ngoku massanas*. These *massana* bring the community together and usually all camp members participate (other than specific *massana* in which only one sex participates). Whilst discourse surrounding Indigenous alcohol use often focusses on disease and cultural loss, these data act as a reminder that it is not alcohol use in itself which is problematic, only that which is harmful to people's health or psychological wellbeing. The effect of alcohol that most concerned Mbendjele people in this study was violence, and interviews indicated harmful community effects may be more pronounced in settled villages than in forest camps. Domestic abuse and violence are widely and consistently mentioned by researchers of transitioning hunter-gatherer societies, and invariably attributed to alcohol use [8, 29, 45]. Therefore, interventions which aim to reduce alcohol-related violence should be a priority to any future public health initiative.

Alcohol use may also perpetuate economic inequalities between Mbendjele people and their Bantu neighbours. Participants reported being paid in alcohol for their labour, as well as spending a substantial proportion of any income, incurring debts and selling/exhanging foraged food to acquire alcohol. Debts were occasionally reported to double in as little as two days, and people reported spending between 10–66% of their most recent income on alcohol. Other ethnographers have reported that Bantu people often provide alcohol for free to BaYaka teenagers and young men to encourage dependence, which can later be used to generate large debts and secure control over their labour [10, 17]. Acquisition of alcohol may therefore be a significant incentive for many Mbendjele people to engage in work outside the forest, whilst cementing their marginalised economic position in the village world.

## Limitations

This study provides novel insights into the patterns and impacts of Mbendjele BaYaka alcohol use; however, it is limited in several ways. The small sample size in the alcohol-biometric analysis (n = 83) increases the likelihood of type II errors. Also, given the conservative estimates, which used lower bounds of %ABV, reported consumption is likely to be an underestimate. Moreover, Mbendjele people are less familiar with our modes of measuring time and quantity, introducing further uncertainty in our estimates which were based on self-reports. Nevertheless, the close correspondence between self-reported consumption levels in our study and direct observations in the Baka study suggest this may not be an issue. Finally, drinking patterns among the Mbendjele are irregular, particularly in forest camps, and determined by highly variable access to vendors, thus repeat measurements are essential to increase data resolution.

## Conclusions

Given the lack of robust data on alcohol use in Indigenous African populations, this study represents an important initial step, and offers guidance for future research. The results provide empirical evidence for ethnographic accounts of dangerous drinking and related medical and

social problems in BaYaka communities undergoing rapid social change; and suggest alcohol consumption is higher among the BaYaka than urban populations in Central Africa.

These studies are essential to produce informed population-specific public health interventions. Particular priorities for future interventions should include reducing alcohol related violence and drinking during pregnancy and breastfeeding; and improving awareness of the physical health consequences of hazardous drinking. Such interventions must consider differences in the prevalence of these issues between settled villages and forest camps as well as within-camp variation according to age and sex. It is also essential that the communities are involved and empowered during the process [1, 6]. Research and interventions related to alcohol use and its effects among Australian Aborigines are substantially more developed and may offer direction for potential approaches, see [46] for review. One recent initiative in Congo is Project Bwanga, which trains traditional Mbendjele healers to run dispensaries and administer primary healthcare; these individuals could also be trained to educate others about the health effects of excessive alcohol use and drinking during pregnancy.

This study also highlights the urgent need for further research, approaches may include:

- Using a diagnostic tool, such as the WHO AUDIT, to assess the prevalence of alcohol use disorders.

- Analysing the ethanol and methanol content of home-brewed beverages, that may be highly variable in composition (we attempted this, however the test kits used did not function correctly, most likely because of the high temperatures and humidity in the forest).

- Assessing the prevalence of diseases closely linked to alcohol, including fetal alcohol spectrum disorders.

- Identifying the specific drinking patterns (e.g. occurrence of heavy episodic drinking) of women who are pregnant and breastfeeding.

- Study of other substance use, particularly glue sniffing.

Harmful drinking in BaYaka communities is intricately tied to the pressures of rapid acculturation, discrimination, sedentarisation, and integration into market economies. Socioeconomic inequality is a fundamental cause of poor health [47]; as such, any interventions which aim to improve health outcomes for BaYaka people must be accompanied by wider changes that address their marginalised position in local and global socioeconomic systems.

## Supporting information

**S1 File. The supporting information file includes English translations of structured interviews; coding categories for interview responses; responses to question about whether alcohol causes problems, separated by camp; and full results tables of all regression analyses.** (DOCX)

**S1 Dataset.** (XLSX)

## Acknowledgments

We thank the Mbendjele for their hospitality and participation; our translators Mindula, Nicolas and Roger for their help in data collection; and Clobite Bouka Biona and Laure Stella Ghoma Linguissi for their help with fieldwork logistics and research permits.

## Author Contributions

**Conceptualization:** Gul Deniz Salali, Nikhil Chaudhary.

**Data curation:** Jessica K. Knight, Nikhil Chaudhary.

**Formal analysis:** Jessica K. Knight, Nikhil Chaudhary.

**Funding acquisition:** Gul Deniz Salali.

**Investigation:** Gul Deniz Salali, Gaurav Sikka, Inez Derkx, Sarai M. Keestra, Nikhil Chaudhary.

**Methodology:** Jessica K. Knight, Gul Deniz Salali, Nikhil Chaudhary.

**Supervision:** Gul Deniz Salali, Nikhil Chaudhary.

**Writing – original draft:** Jessica K. Knight, Nikhil Chaudhary.

**Writing – review & editing:** Gul Deniz Salali, Gaurav Sikka, Inez Derkx, Sarai M. Keestra.

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
