## [Decision Letter · Decision Letter 0]

11 Jun 2021

PONE-D-21-13970

Quantifying patterns of alcohol consumption and its effects on health and wellbeing among BaYaka hunter-gatherers: A mixed-methods cross-sectional study

PLOS ONE

Dear Dr. Chaudhary,

Thank you for submitting your manuscript to PLOS ONE. After careful consideration, we feel that it has merit but does not fully meet PLOS ONE’s publication criteria as it currently stands. Therefore, we invite you to submit a revised version of the manuscript that addresses the points raised during the review process.

This ethnographic study provides very interesting findings from a study which examined quantitative patterns of alcohol consumption and their health and social impact on Congolese Mbendjele BaYaka (hunter-gatherers) population undergoing socioeconomic transition. Only one previous study appears to have been published, which provided such quantitative data. 83 adults provided frequency and quantity of alcohol consumption data and underwent biometric assessments; 56 responded to structured interviews. Almost half engaged in heavy episodic drinking, which varied by sex and age.

I would like to recommend that the authors insert a small map of Africa/Congo showing where the three Mbendjele communities are located (insert around Line 128).

Both reviewers and I agree that the paper is very well-written and describes a well-conducted, challenging study. Both also recommend inclusion of a sample characteristics table with descriptive and quantitative information about demographic and biometric data collected, which would be organized showing mean values with for each of the three communities described.

Reviewer 1 raised a very important question: in calculating alcohol use, did you include abstainers or only alcohol users?  Since the number of abstainers can vary in any population, their inclusion in describing alcohol use in a particular population can greatly impact on the amount. This should be based on alcohol users only.

I do not agree with Reviewer 2 that you need to mention the lack of a structured interview as a limitation at the end of the paper.  Moreover, the AUDIT should not substitute for quantitative questions, like the ones used in this paper. The AUDIT-C includes quantitative data and is fine. You might also recommend a timeline follow-back type interview (e.g., Jacobson et al., Pediatrics, 2002). But I do not think that in this paper it is necessary to convert to oz absolute alcohol based on the exact amount of alcohol reported, as recommended by Reviewer 2. This would be interesting and might generate more associations with the outcome variables, but I will leave this up to the authors whether to do so or not.

Lines 72-74 The following sentence is confusing: “An exception is a study that found raised serum gamma-glutamyl transferase—a biomarker of heavy drinking (Peterson 2004)—in 30% of men and 11% of women in San communities, which were demonstrated to contribute to thiamine deficiency in the sample (van der Westhuyzen et al. 1987).” Was the study conducted in 2004 or 1987? Was the reference to Peterson a reference to the van der Westhuzuyzen study or a separate study? If the latter, the first reference is not necessary.

Line 151 Who collected the physiological variables? What was their training—doctors, nurses, etc.?

Line 187 In 1-2 sentences, briefly indicate how estimated ages were generated, since this will be of interest and not readily available to the readers.

Lines 243-4 Do the authors have any idea why “Drinking during breastfeeding was more than five times as common in the camp of Longa than the settlement of Sembola?

Lines 348-9 It is of much public health interest that “Drinking while breastfeeding was significantly less common in Sembola than Longa indicating that proximity to a hospital and accessible public health information may influence drinking behaviour.” However, I would change the word “indicating” to “suggesting”.

We look forward to receiving your revised manuscript.

Kind regards,

Sandra Jacobson

Academic Editor

PLOS ONE

2. During our internal checks, the in-house editorial staff noted that you conducted research or obtained samples in another country. Please check the relevant national regulations and laws applying to foreign researchers and state whether you obtained the required permits and approvals. Please address this in your ethics statement in both the manuscript and submission information. In addition, please ensure that you have suitably acknowledged the contributions of any local collaborators involved in this work in your authorship list and/or Acknowledgements. Authorship criteria is based on the International Committee of Medical Journal Editors (ICMJE) Uniform Requirements for Manuscripts Submitted to Biomedical Journals - for further information please see here: https://journals.plos.org/plosone/s/authorship.

3. Please include a copy of the interview guide used in the study, in both the original language and English, as Supporting Information, or include a citation if it has been published previously.

Reviewers' comments:

Reviewer's Responses to Questions

**Comments to the Author**

1. Is the manuscript technically sound, and do the data support the conclusions?

Reviewer #1: Partly

Reviewer #2: Yes

2. Has the statistical analysis been performed appropriately and rigorously? 

Reviewer #1: Yes

Reviewer #2: No

3. Have the authors made all data underlying the findings in their manuscript fully available?

Reviewer #1: Yes

Reviewer #2: No

4. Is the manuscript presented in an intelligible fashion and written in standard English?

Reviewer #1: Yes

Reviewer #2: Yes

5. Review Comments to the Author

Reviewer #1: This study examined alcohol use and its impact on health and social problems in Mbendjele BaYaka, an indigenous hunter-gatherer community undergoing socioeconomic transition. The authors reported that almost half of participants reported both hazardous regular consumption and heavy episodic drinking. Weekly alcohol consumption predicted systolic blood pressure and the likelihood of diarrhoea. High rates of drinking during pregnancy and breastfeeding were also reported. These data indicate a need for targeted public health interventions to reduce the hazardous consumption of alcohol and increase awareness of the harmful effects of drinking while pregnant. The article is well-written and the conclusions are supported by the data being reported. However, I have some questions about how hazardous volume and HED were defined.

The WHO defines hazardous alcohol consumption as “averaging more than two drinks – 24 grams of ethanol – per day”. HED is defined as “ drinking about five or more drinks – roughly 60 grams of ethanol”. Therefore, WHO defines a drink as 12 grams of ethanol. From the reported types of beverage consumed the authors calculated a weighted mean ethanol content of 29.4%, and a cup is defined as 100 ml. From this I calculate that 1 Mbendjele cup contains roughly 23 grams of ethanol on average, or just under 2 standard drinks. This would indicate that hazardous drinking should be defined as 7.3 Mbendjele cups of alcohol per week, and HED as 2.6 Mbendjele cups per occasion. Why are the authors instead using more lenient definitions of 5.7 and 2.0 Mbendjele cups instead?

Lines 176 – 178 – These should also be defined in terms of ‘standard drinks’. The WHO defines hazardous alcohol consumption as “averaging more than two drinks – 24 grams of ethanol – per day”. HED is defined as “ drinking about five or more drinks – roughly 60 grams of ethanol”.

Lines 205-208 – Do the means reported here include abstainers? If so, the authors should also present these data excluding abstainers, so we can get an accurate picture of how the “drinkers” drink. A true sample characteristics table showing age, sex, and quantity and frequency of alcohol consumption for the whole sample, and by hazardous drinking and HED groups is needed. Additionally, reporting the alcohol volume data in standard drinks (1 drink = 12g ethanol according to WHO) in addition to Mbendjele drinks would help facilitate readers in comparing with other populations.

Line 212 – What is “apollon” ? It is not defined in Table 1.

Lines 221-222 – Is this statistically different?

Table 2 – The rows with Total are not presenting any new information and are not needed.

Lines 337-353 -- The reported rate of pregnancy drinking is very high, and it seems like the awareness of potential harms of drinking is lacking. Are traditional public health interventions, like pamphlets and interviewing during prenatal visits effective in hunter-gatherer populations like the Mbendjele? Why do the authors think that proximity to a hospital and public health information would reduce drinking while breastfeeding, but not during pregnancy in the Sembola camp?

Reviewer #2: This is a mixed-methods, cross-sectional study of 3 BaYaka hunter-gatherer communities in Congo aimed at quantifying alcohol use, evaluating potential relations between alcohol use and health outcomes, and describing perceptions/beliefs/practices regarding alcohol use. Strengths of the study include the methodology, the use of 3 communities, and the new information regarding alcohol use in this community. The manuscript is clearly and concisely written. The examination of potential quadratic relations (rather than assuming relations were linear) is commendable.

One major methodological issue is that weighted alcohol content based on the alcohol types consumed was used to calculate dose per research participant. Since the authors have volume and type of beverage for each individual, dose/individual should be calculated based on the exact volume and beverages that each individual reported. Copious research shows that the % absolute alcohol by volume in a given beverage may drastically affect the quantity consumed, as individuals often drink with the aim of achieving a certain feeling or “high,” which is accomplished by consuming a certain dose of absolute alcohol rather than a set beverage volume. This will require recalculating alcohol use values for all participants and rerunning the analyses, but doing so will greatly strengthen this manuscript.

The results should include a table and description of the descriptive, demographic, and biometric data collected, with mean values/group %s for each of the 3 communities studied. Such a description will help put into context the subsequent results.

Tables S2-S4 would be interesting to have in the main manuscript (i.e., not just in the Supplemental material).

Line 212: were 82 types of beverage reported or did 82 respondents report beverages, including the 6 types described?

In table 2 and elsewhere, %s should be given as n (%).

Were there interesting differences in findings between the 3 communities examined? If power is an issue, it may be worth comparing Sembola to Longa/Njoki combined.

Do the authors have any data regarding the frequency and/or dose of alcohol consumption during pregnancy and breastfeeding? It would be interesting to assess, even if just descriptively, whether women reduce drinking during pregnancy/lactation and to provide any estimates of the prevalence of heavy episodic drinking, as that pattern is the most teratogenic.

Lines 381-401: Although drinking during pregnancy/lactation was discussed in more depth above, this should be mentioned as a negative health outcome here as well.

The gold standard for alcohol interviews is the timeline-followback interview, in which respondents are about their drinking during the previous 2 weeks on a day-by-day basis. The lack of use of this structured interview should be mentioned as a limitation.

Lastly, in the data availability statement, the authors should describe how data will be made available.

6. PLOS authors have the option to publish the peer review history of their article (what does this mean?). If published, this will include your full peer review and any attached files.

Reviewer #1: No

Reviewer #2: No

---

## [Author Response · Author response to Decision Letter 0]

28 Jul 2021

Please see uploaded document in order to see italics and text colour.

Dear Editor and Reviewers

We thank you for your consideration of our manuscript and the insightful and useful comments and suggestions that have greatly improved the submitted revised manuscript. Below we respond to all comments/ queries (red italics) made by the editor and both reviewers and provide information/quotes (italics) highlighting the corresponding changes made to the manuscript. Line numbers specified here refer to the “Revised Manuscript with Track Changes” document.

Comments from the editor

1. I would like to recommend that the authors insert a small map of Africa/Congo showing where the three Mbendjele communities are located (insert around Line 128).

This map has now been included, see L133.

2. Both reviewers and I agree that the paper is very well-written and describes a well-conducted, challenging study. Both also recommend inclusion of a sample characteristics table with descriptive and quantitative information about demographic and biometric data collected, which would be organized showing mean values with for each of the three communities described.

We have now included a sample characteristics table (Table 2), see L184. This table provides demographic data for each of the communities for each phase of the study (biometric and interview). 

We do not include descriptive statistics relating to the biometric data itself, e.g. mean white blood cell count for the entire sample etc. These data are being prepared for another publication and also ought to only be interpreted with detailed context such as a break down of measurements across age and sex groups.

3. Reviewer 1 raised a very important question: in calculating alcohol use, did you include abstainers or only alcohol users? Since the number of abstainers can vary in any population, their inclusion in describing alcohol use in a particular population can greatly impact on the amount. This should be based on alcohol users only.

Thank you for raising this important point. We now include calculations for both the “drinking sample” and the entire sample, see lines 246-250.

4. I do not agree with Reviewer 2 that you need to mention the lack of a structured interview as a limitation at the end of the paper. Moreover, the AUDIT should not substitute for quantitative questions, like the ones used in this paper. The AUDIT-C includes quantitative data and is fine. You might also recommend a timeline follow-back type interview (e.g., Jacobson et al., Pediatrics, 2002). But I do not think that in this paper it is necessary to convert to oz absolute alcohol based on the exact amount of alcohol reported, as recommended by Reviewer 2. This would be interesting and might generate more associations with the outcome variables, but I will leave this up to the authors whether to do so or not.

We now include the following explanation (L156) as to why we did not use timeline follow-back interviews:

We chose the above questions rather than timeline follow-back methods in which participants recount their consumption over the last two weeks. This is because Mbendjele alcohol consumption is not consistent across time, rather it is determined by access, ability to purchase and the timing of ritual events. Therefore, asking a general question rather than focussing on the previous two weeks is likely to elicit average habits and provide a more representative depiction of consumption patterns.

5. Lines 72-74 The following sentence is confusing: “An exception is a study that found raised serum gamma-glutamyl transferase—a biomarker of heavy drinking (Peterson 2004)—in 30% of men and 11% of women in San communities, which were demonstrated to contribute to thiamine deficiency in the sample (van der Westhuyzen et al. 1987).” Was the study conducted in 2004 or 1987? Was the reference to Peterson a reference to the van der Westhuzuyzen study or a separate study? If the latter, the first reference is not necessary.

We have now removed the Peterson reference.

6. Line 151 Who collected the physiological variables? What was their training—doctors, nurses, etc.?

These were collected by an NHS physician, we now specify this, see L164.

7. Line 187 In 1-2 sentences, briefly indicate how estimated ages were generated, since this will be of interest and not readily available to the readers.

We now include the following description, see L219-224:

Estimated ages had already been produced for a large sample of this study population by creating relative age lists with the community and assigning age ranges for individuals based on dental development, sibling birth orders and triangulating their birth years with known events such as the construction of a logging road. Final age estimates were then obtained by integrating this information using a Gibbs sampling framework. Detailed information and validation of this method can be found in Diekmann et al. 2017.

8. Lines 243-4 Do the authors have any idea why “Drinking during breastfeeding was more than five times as common in the camp of Longa than the settlement of Sembola?

On L397, we speculate that:

Drinking while breastfeeding was significantly less common in Sembola than Longa suggesting that proximity to a hospital and accessible public health information may influence drinking behaviour

9. Lines 348-9 It is of much public health interest that “Drinking while breastfeeding was significantly less common in Sembola than Longa indicating that proximity to a hospital and accessible public health information may influence drinking behaviour.” However, I would change the word “indicating” to “suggesting”.

We have now changed this.

Comments from Reviewer 1

1. The WHO defines hazardous alcohol consumption as “averaging more than two drinks – 24 grams of ethanol – per day”. HED is defined as “ drinking about five or more drinks – roughly 60 grams of ethanol”. Therefore, WHO defines a drink as 12 grams of ethanol. From the reported types of beverage consumed the authors calculated a weighted mean ethanol content of 29.4%, and a cup is defined as 100 ml. From this I calculate that 1 Mbendjele cup contains roughly 23 grams of ethanol on average, or just under 2 standard drinks. This would indicate that hazardous drinking should be defined as 7.3 Mbendjele cups of alcohol per week, and HED as 2.6 Mbendjele cups per occasion. Why are the authors instead using more lenient definitions of 5.7 and 2.0 Mbendjele cups instead?

We thank the reviewer for highlighting this extremely important issue. It seems the conversion of ml to grams of ethanol was not executed in our code due to an error. We have now corrected this, and have adjusted the prevalence rates of HED and hazardous drinking in Table 3, chi-sq tests of sex-based differences in prevalence, as well as the indicator line in Figure 2; see L256-272.

2. Lines 176 – 178 – These should also be defined in terms of ‘standard drinks’. The WHO defines hazardous alcohol consumption as “averaging more than two drinks – 24 grams of ethanol – per day”. HED is defined as “ drinking about five or more drinks – roughly 60 grams of ethanol”.

We have now included these definitions, see L192-195.

3. Lines 205-208 – Do the means reported here include abstainers? If so, the authors should also present these data excluding abstainers, so we can get an accurate picture of how the “drinkers” drink. A true sample characteristics table showing age, sex, and quantity and frequency of alcohol consumption for the whole sample, and by hazardous drinking and HED groups is needed. Additionally, reporting the alcohol volume data in standard drinks (1 drink = 12g ethanol according to WHO) in addition to Mbendjele drinks would help facilitate readers in comparing with other populations.

We have now included “drinkers” as well as the entire sample, see L246-250.

We have also included the sample characteristics requested in Table 3.

We have not included a supplemental conversion to “standard drinks” to avoid confusion among readers as to whether subsequent in-text references to drinks/cups refers to Mbendjele cups or “standard drinks”.

4. Line 212 – What is “apollon” ? It is not defined in Table 1.

We now specify this is a “gin brand”, see L255.

5. Lines 221-222 – Is this statistically different?

We now report results demonstrating these differences in age are statistically significant, see L266-268:

Notably the mean age of participants who did not drink was significantly younger, by more than 20 years, than both those who drank at hazardous levels (W=11.5, p=0.001) and those that engaged in heavy episodic drinking (W=15, p=0.001), see Table 3 for summary.

6. Table 2 – The rows with Total are not presenting any new information and are not needed.

These have now been removed.

7. Lines 337-353 -- The reported rate of pregnancy drinking is very high, and it seems like the awareness of potential harms of drinking is lacking. Are traditional public health interventions, like pamphlets and interviewing during prenatal visits effective in hunter-gatherer populations like the Mbendjele? Why do the authors think that proximity to a hospital and public health information would reduce drinking while breastfeeding, but not during pregnancy in the Sembola camp?

We address these questions in L397-407:

Drinking while breastfeeding was significantly less common in Sembola than Longa suggesting that proximity to a hospital and accessible public health information may influence drinking behaviour. However, it is unclear then why the prevalence of drinking during pregnancy is very similar in both camps. One possibility is that understanding the effects of alcohol on the baby from breastfeeding may be more intuitive because the baby is visibly ingesting fluid, whereas nutrient transfer via the placenta during pregnancy is unobservable. Despite key structural determinants of alcohol use, brief communication of the risks of alcohol can be an effective, low-cost and easy-to-administer intervention, and has been utilised successfully in urban Congolese settings (Williams et al. 2014). Our findings suggests that such interventions may be beneficial for Mbendjele people, though their effectiveness among hunter-gatherer societies remains to be demonstrated.

Comments from Reviewer 2

1. One major methodological issue is that weighted alcohol content based on the alcohol types consumed was used to calculate dose per research participant. Since the authors have volume and type of beverage for each individual, dose/individual should be calculated based on the exact volume and beverages that each individual reported. Copious research shows that the % absolute alcohol by volume in a given beverage may drastically affect the quantity consumed, as individuals often drink with the aim of achieving a certain feeling or “high,” which is accomplished by consuming a certain dose of absolute alcohol rather than a set beverage volume. This will require recalculating alcohol use values for all participants and rerunning the analyses, but doing so will greatly strengthen this manuscript.

We understand reviewer 2’s comment and now explain in the manuscript why the suggested approach was not implemented, see L196-211.

The frequencies of different types of alcohol reported were multiplied by the lowest previously reported percentage alcohol by volume (%ABV) to estimate a conservative weighted mean ethanol content. We applied this weighted mean ethanol content per drink to the whole sample rather than calculating ethanol content per drink for each individual based on the specific drink reported by each participant. This is because the Mbendjele drink together in groups and will usually drink the same drink as the rest of the party on any given occasion. Whilst it may be unrelatable in industrialised cultures, our understanding is that individuals do not decide whether to drink or not on a given occasion based on which drinks are available, choice of drink is a secondary decision and usually determined by availability. It is likely the question was interpreted as which drink would the participant choose if all drinks were available, and it is certainly not the case that individuals only consume the drink they reported in their response. Estimating average ethanol content per drink for each individual based on each participant’s specific drinking history would be ideal if it were possible to accurately obtain this data. However, we do not believe this would change our results very much since 90% of responses listed either lotoko, pastis or apollon, which are all spirits with similar %ABV (~40%), and the other drinks listed are less commonly available.

2. The results should include a table and description of the descriptive, demographic, and biometric data collected, with mean values/group %s for each of the 3 communities studied. Such a description will help put into context the subsequent results.

See response to editor comment 2.

3. Tables S2-S4 would be interesting to have in the main manuscript (i.e., not just in the Supplemental material).

We have not included these tables in the manuscript because we feel having 6 tables and 5 figures would be too crowded and too much information for the main text. Hence we have decided to keep these in the supplementary, but are willing to change this if considered essential.

4. Line 212: were 82 types of beverage reported or did 82 respondents report beverages, including the 6 types described?

We now clarify this, see L254:

Eighty-two responses to “which type of alcohol do you drink” were given

5. In table 2 and elsewhere, %s should be given as n (%).

We have now included n(%) format in all tables.

6. Were there interesting differences in findings between the 3 communities examined? If power is an issue, it may be worth comparing Sembola to Longa/Njoki combined.

We allude to such differences in the results section of the main text, e.g.:

L289: Drinking during breastfeeding was more than five times as common in the camp of Longa than the settlement of Sembola (chi-squared=20.7, p<0.0001),

L309: though some people in Sembola earned money from other sources e.g. work for the forestry company.

L319: When interviewees were asked whether they thought alcohol causes problems, violence (including domestic violence) was the most frequent response (44.4% of responses), and reported significantly more often in Sembola than Longa (chi-squared=5.97, p=0.015).

7. Do the authors have any data regarding the frequency and/or dose of alcohol consumption during pregnancy and breastfeeding? It would be interesting to assess, even if just descriptively, whether women reduce drinking during pregnancy/lactation and to provide any estimates of the prevalence of heavy episodic drinking, as that pattern is the most teratogenic.

Unfortunately, we do not have this data; we now include this as a suggestion for future work, see L494:

Identifying the specific drinking patterns (e.g. occurrence of heavy episodic drinking) of women who are pregnant and breastfeeding.

Lines 381-401: Although drinking during pregnancy/lactation was discussed in more depth above, this should be mentioned as a negative health outcome here as well.

We do not mention any health outcomes there because that section is entitled Social and economic impact.

The gold standard for alcohol interviews is the timeline-followback interview, in which respondents are about their drinking during the previous 2 weeks on a day-by-day basis. The lack of use of this structured interview should be mentioned as a limitation.

See response to editor comment 4.

Lastly, in the data availability statement, the authors should describe how data will be made available.

The complete data availability statement will be available if the manuscript is published.

Once again, we thank the editor and both reviewers for their time and helpful comments. We hope you are satisfied with our response and amendments, and look forward to hearing from you.

Yours sincerely 

Dr Nikhil Chaudhary

---

## [Decision Letter · Decision Letter 1]

27 Sep 2021

Quantifying patterns of alcohol consumption and its effects on health and wellbeing among BaYaka hunter-gatherers: A mixed-methods cross-sectional study

PONE-D-21-13970R1

Dear Dr. Chaudhary,

We are very pleased to inform you that you have addressed all of our comments and that your manuscript has been judged scientifically suitable for publication and will be formally accepted for publication once it meets all outstanding technical requirements.

Within one week, you will receive an e-mail detailing the required amendments. When these have been addressed, you will receive a formal acceptance letter and your manuscript will be scheduled for publication.

If your institution or institutions have a press office, please notify them about your upcoming paper to help maximize its impact. If they will be preparing press materials, please inform our press team as soon as possible -- no later than 48 hours after receiving the formal acceptance. Your manuscript will remain under strict press embargo until 2 pm Eastern Time on the date of publication. For more information, please contact onepress@plos.org.

Again, congratulations on this fine study.

Kind regards,

Sandra Jacobson

Academic Editor

PLOS ONE

Additional Editor Comments (optional):

The authors have done an excellent job at addressing all of the comments and questions raised in the review, including addition of a map showing where the three Mbendjele communities are located. This is a challenging study, and I congratulate the authors on their ability to conduct the research and to adapt the alcohol measures and apply them in such a way as to generate their very interesting findings. I anticipate that this paper will be of much interest to many researchers in the field.

Reviewers' comments:

Reviewer's Responses to Questions

**Comments to the Author**

1. If the authors have adequately addressed your comments raised in a previous round of review and you feel that this manuscript is now acceptable for publication, you may indicate that here to bypass the “Comments to the Author” section, enter your conflict of interest statement in the “Confidential to Editor” section, and submit your "Accept" recommendation.

Reviewer #2: All comments have been addressed

2. Is the manuscript technically sound, and do the data support the conclusions?

Reviewer #2: Yes

3. Has the statistical analysis been performed appropriately and rigorously? 

Reviewer #2: Yes

4. Have the authors made all data underlying the findings in their manuscript fully available?

Reviewer #2: No

5. Is the manuscript presented in an intelligible fashion and written in standard English?

Reviewer #2: Yes

6. Review Comments to the Author

Reviewer #2: The authors have done an excellent job of responding to the review critiques, thus strengthening this excellent and novel manuscript.

7. PLOS authors have the option to publish the peer review history of their article (what does this mean?). If published, this will include your full peer review and any attached files.

Reviewer #2: No

---

## [Editor Report · Acceptance letter]

1 Oct 2021

PONE-D-21-13970R1 

Quantifying patterns of alcohol consumption and its effects on health and wellbeing among BaYaka hunter-gatherers: a mixed-methods cross-sectional study 

Dear Dr. Chaudhary:

I'm pleased to inform you that your manuscript has been deemed suitable for publication in PLOS ONE. Congratulations! Your manuscript is now with our production department. 

Kind regards, 

on behalf of

Dr. Sandra Jacobson 

Academic Editor

PLOS ONE